# Correlation of Molecular Status with Preoperative Olfactory Function in Olfactory Groove Meningioma

**DOI:** 10.3390/cancers16081595

**Published:** 2024-04-22

**Authors:** Dino Podlesek, Friederike Beyer, Majd Alkhatib, Dirk Daubner, Mido Max Hijazi, Jerry Hadi Juratli, Susanne Weise, Ilker Y. Eyüpoglu, Gabriele Schackert, Tareq A. Juratli, Thomas Hummel

**Affiliations:** 1Department of Neurosurgery, Faculty of Medicine, Carl Gustav Carus University Hospital, Technische Universität Dresden, 01307 Dresden, Germany; 2Institute of Diagnostic and Interventional Neuroradiology, Carl Gustav Carus University Hospital, Technische Universität Dresden, 01307 Dresden, Germany; 3Department of Otorhinolaryngology, Smell & Taste Clinic, Carl Gustav Carus University Hospital, Technische Universität Dresden, 01307 Dresden, Germany

**Keywords:** olfactory groove meningiomas, olfactory impairment, *AKT1* and *SMO* mutation, preoperative phenotypic characteristics of OGM

## Abstract

**Simple Summary:**

This study investigates the relationship between genomic alterations and preoperative olfactory function in patients with olfactory groove meningioma (OGM), often associated with olfactory impairment. Utilizing next-generation sequencing on 22 individuals with OGM, the research identified mutations with *SMO/SUFU*, with *AKT1*, and as wild type. The presence of planum sphenoidale hyperostosis (PSH) correlated with significant variations in tumour morphology and negatively impacted olfactory function, affecting odour threshold, discrimination, identification, and overall olfactory performance. Additionally, perifocal oedema was linked to decreased olfactory performance. The study also found that age over 65 and female gender were associated with reduced olfactory capabilities. Despite these findings, no direct link between olfactory impairment and tumour mutations was established, possibly due to the limited sample size. The research suggests further investigation with a larger participant group in order to better understand the impact of OGM driver mutations on olfactory function.

**Abstract:**

Purpose: The study aims to examine the possible correlation between genomic alterations and preoperative olfactory function in patients with olfactory groove meningioma (OGM), due to the frequent presence of olfactory impairment. Methods: We utilised next-generation sequencing to analyse samples from 22 individuals with OGM in order to detect driver mutations. Tumour morphology was assessed using preoperative imaging, whereas olfactory function was examined using Sniffin’ Sticks. Results: In a study of 22 OGM patients, mutations were as follows: 10 with *SMO/SUFU*, 7 with *AKT1*, and 5 as wild type. Planum sphenoidale hyperostosis (PSH) was present in 75% of patients, showing significant variation by mutation (*p* = 0.048). Tumour volumes, averaging 25 cm^3^, significantly differed among groups. PSH negatively impacted olfaction, notably affecting odour threshold, discrimination, identification, and global olfactory performance score (TDI) (*p* values ranging from <0.001 to 0.003). Perifocal oedema was associated with lower TDI (*p* = 0.009) and altered threshold scores (*p* = 0.038). Age over 65 and female gender were linked to lower thresholds and discrimination scores (*p* = 0.037 and *p* = 0.019). Conclusion: The study highlights PSH and perifocal oedema’s significant effect on olfactory function in OGM patients but finds no link between olfactory impairment and tumour mutations, possibly due to the small sample size. This suggests that age and gender affect olfactory impairment. Additional research with a larger group of participants is needed to explore the impact of OGM driver mutations on olfactory performance.

## 1. Background

Meningiomas are the most prevalent type of intracranial tumour, accounting for up to 37% of cases [1]. Olfactory groove meningiomas (OGM), which represent about 10% of all intracranial meningiomas, arise from the cribriform plate and frontosphenoidal suture [2]. While OGMs are typically benign, their location can lead to significant comorbidities, resulting in symptoms that include headache, cognitive decline, diminished olfactory performance, and visual field impairment [3].

Olfactory deficits have been noted in 72% of OGM patients, frequently leading to a diminished quality of life [4]. Despite this, olfactory dysfunction seldom contributes to the diagnosis of OGM, highlighting the necessity of incorporating validated psychophysical tests into the diagnostic process [5]. The preservation of olfactory function during OGM surgery is challenging, as it depends on tumour location and proximity to olfactory structures [6,7]. Preoperative normosmia and tumour volume are crucial predictors for postoperative preservation of olfactory performance [8].

The influence of genetic mutations on olfactory function in OGM remains unknown, despite the established impact of tumour genomics on factors such as location, recurrence potential, imaging features, and patient outcomes [9]. Therefore, exploring molecular alterations and their association with psychophysical assessments could offer significant insights. Common mutations in OGM include those in genes such as *AKT1*, *SMO*, and *PIK3CA* [10,11]. *AKT1* mutations are involved in cell survival pathways, while *SMO* mutations have a key role in the hedgehog signalling pathway that guides cell proliferation. Additionally, *PIK3CA* mutations play a part in the *PI3K*/Akt signalling pathway which regulates cell growth and survival.

While previous studies have focused on the relationship between mutational status, tumour histology, recurrence, and outcomes in meningiomas [12,13], our study aims to delineate subgroups of OGM patients based on the impact of mutational status and image morphology on preoperative olfactory performance. We hypothesize that the mutational status of OGM affects preoperative olfactory performance and is associated with distinct image characteristics.

## 2. Methods

### 2.1. Study Design and Methods

Written informed consent was obtained from all participating patients. Inclusion criteria consisted of accessible genetic tissue analysis of olfactory groove meningioma (OGM).

### 2.2. Olfactory Performance Testing

The preoperative evaluation of the patient’s olfactory function was performed using the Sniffin’ Sticks olfactory test battery upon admission for tumour removal surgery.

Sniffin’ Sticks olfactory test is a widely used, validated psychophysical assessment of olfactory performance. Pen-like odour dispensing devices (sticks) are held in front of the patient’s nose which comprise tests for odour threshold, odour discrimination, and odour identification [14,15]. The olfactory threshold was determined for phenylethanol in 16 dilution steps. Three sets of pens were presented, one of which contained the diluted odour, while the others were odourless blanks. Subjects were required to identify the odorous pen in a three-alternative forced-choice paradigm (3-AFC). Using a staircase paradigm, two consecutive correct identifications prompted the presentation of the next higher dilution, while one incorrect identification triggered the return to a lower dilution. After seven turning points, the threshold was calculated as the mean of the dilution steps of the last four turning points. Odour discrimination was also assessed in a 3-AFC design using 16 triplets, with 2 pens having the same odorant and 1 pen having a different odour, the task of which is to identify the pen with the different odour. Odour identification was assessed in a 4-AFC design, in which subjects were required to name 16 odours from 4 alternatives presented with each odour. The final TDI score was the sum of the scores for the threshold, discrimination, and identification subtests with a range between 1 and 48 points, representing an overall score of olfactory function [16]. Patients were categorized as normosmic (TDI score > 30.5), anosmic (TDI score ≤ 16.5), or hyposmic (scores between normosmia and anosmia).

### 2.3. Pre- and Postoperative Imaging

Baseline magnetic resonance tomography scans included T1-/T2-weighted, contrast-enhanced T1-weighted, and fluid attenuation inversion recovery (FLAIR) imaging. Additionally, the preoperative magnetization prepared rapid acquisition gradient echo (MP-RAGE) sequence was available for most patients (1.5T; Sonata; Siemens, Erlangen, Germany). Cranial computer tomography (CT) scan or MR images were used to assess PSH, perifocal tumour oedema, and osseous infiltration by an independent neuroradiologist (D.D.). Tumour volume estimation utilized maximum tumour diameters measured in three orthogonal directions Although there are several approaches to determine tumour volume, we utilized the (V = (4/3) · π·r3) formula, as OGMs typically have a round or oval shape [16,17,18,19,20].

### 2.4. Tumour Sequencing

The tumour area was marked on a haematoxylin–eosin (H&E) stained slide by a board-certified pathologist, and corresponding tumour areas were macrodissected from the FFPE block. Genomic DNA was extracted from the collected tumour material using the QIAamp DNA Mini Kit (Qiagen, Hilden, Germany). DNA concentration was quantified using the Qubit™ dsDNA BR Assay (Life Technologies Europe, Bleiswijk, The Netherlands).

Analogous to our previously described methodology [21], we used a custom-designed amplikon panel from Qiagen to amplify mutation hotspots of genes (in case of point mutation) or whole genes (in case of loss of function) of *AKT1*, *ATRX*, *BRAF*, *CDKN2A*, *CIC*, *DAXX*, *EGFR*, *GNA11*, *GNAQ*, *H3F2A*, *H3F3B*, *IDH1*, *IDH2*, *KDM6A*, *KLF4, NF1*, *NF2*, *PIK3CA*, *PIK3R1*, *POLR2A, PTEN*, *SMARCB1*, *SMO*, *STAG2*, *SUFU*, *TP53*, *TRAF7*, and *TERTp*. Briefly, amplification was performed according to the protocol “QIAseq targeted DNA panel, May 2017” (Qiagen), followed by paired-end next-generation sequencing (2 × 200 bp) using the Illumina MiSeq platform (Illumina, San Diego, CA, USA). Sequences were then analysed with the Qiagen CLC Genomics Workbench (version 21.0.3) using HG19 as a reference genome and a customized analysis algorithm (coverage  ≥ 500, allele frequency  ≥ 5%).

### 2.5. Statistical Analysis

SPSS software (IBM^®^ SPSS^®^, version 28, Armonk, NY, USA) was used for statistical analyses. Pearson’s Chi-square test was used to examine differences between groups. The dependence between the mutational status and OGM volume was explored by Fisher’s exact test. To point out the difference between olfactory performance and mutational group status, the Kruskal–Wallis H test was used. The correlation between different mutational groups with corresponding OGM volume was expressed by the Mann–Whitney U test. Regression analysis was utilized to estimate the effect of various supplementary variables and to determine the strength of predictors on the results of olfactory subtests. Furthermore, *AKT1*, *SMO/SUFU*, and “Other” mutational groups were compared with the wild type (non-mutated) group. The level of significance was set at *p* < 0.05.

## 3. Results

The study included 22 patients (16 females and 6 males) with a median age of 64 years (range, 46–87) at first diagnosis. The median age of female and male patients was 67 years (range 46–75) and 65 years (range 57–86), respectively. In the following analyses, the study patients were divided into two groups based on their age: ≤65 years (*n* = 9) and >65 years (n = 13) (Table 1). Regarding the driver mutations, *SMO/SUFU* mutations were detected in 10 (46%) and *AKT1* mutations in 7 (32%) patients, while 5 patients (23%) were gathered in a group as *AKT1-/SMO*-wild type (WT). The latter group of meningioma harboured *PIK3CA*, *TRAF7-*, *POLR2A-*, and *NF1* OGM mutations. The median tumour volume was 25 cm^3^ (range: 2–48 cm^3^). Given the limited number of patients, we divided our study cohort into two groups, based on the median OGM volume of 25 cm^3^.

For the radiographic evaluation, we investigated various parameters, including ethmoidal cell infiltration, PSH, osseous enhancement, and the presence of perifocal oedema. Interestingly, a statistically significant association was seen between the mutational status and the presence of PSH (*p* = 0.048). In all patients within the WT group, a PSH was observed. Conversely, in *AKT1*–mutant OGM patients, PSH was documented in 57% of cases.

Table 1 summarizes demographic data, genetic mutations, tumour histology and WHO grade in correlation with the preoperative olfactory function and radiological findings.

The preoperative olfactory threshold, discrimination, and identification scores were available in 21 patients with OGM, while the TDI was available in 22 patients.

The median olfactory threshold score for all patients was 1 (range: 1–10). The only significant difference in olfactory threshold scores was found between patients with perifocal oedema and those without (*p* = 0.038). Patients with perifocal oedema had a higher median olfactory threshold score of 4 (range: 1–10), while those without perifocal oedema had a median score of 1 (range: 0–4). The median olfactory discrimination score was 7 (range 4–13). No significant differences have been detected between the groups. The median olfactory identification score for all patients was 6 (range: 2–16). Notably, the median olfactory identification score of patients without PSH was 14 (*n* = 4; range: 7–16) (*p* = 0.006), whereas patients with a PSH had a lower median score of 4 (*n* = 15; range: 2–11) (*p* = 0.006). TDI showed a median of 13 and a range of 6 to 35. 

Table 2 demonstrates the association of patients’ demographics with tumour characteristics and olfactory subtests.

Through multivariate regression analysis, various parameters were identified as independent predictive factors for preoperative olfactory subtests. The predictors PSH (*p* = 0.001), osseous enhancement (*p* = 0.016), and age (*p* = 0.037) had a statistically significant negative correlation with odour threshold. The presence of PSH and osseous enhancement correlate with a deleterious impact on the threshold, as does an increase in age. Overall, the chosen independent variables together offer a strong explanation for the variation in the threshold, as indicated by the high R^2^ value.

Gender and the presence of PSH were significant predictors of performance on the odour discrimination subtest before the surgical procedure. Furthermore, the coefficient of −2.672 indicates that, compared with women, men generally exhibited odour discrimination scores that were approximately 2.672 units higher, on average (*p* = 0.019). The coefficient for PSH was −3.078, signifying that the presence of PSH was linked to a reduction of about 3.078 units in the odour discrimination test, as opposed to the absence of PSH (*p* = 0.008).

Decreased odour identification scores were associated with the independent prognostic factors PSH (*p* < 0.001) and perifocal oedema (*p* = 0.018).

The presence of PSH (*p* = 0.003) and perifocal oedema in OGM patients (*p* = 0.009) had a significant negative impact on cumulative TDI scores, meaning that individuals with these conditions tended to have lower overall olfactory performance scores.

Table 3 displays the outcomes of multivariate linear regression analysis, where several models investigate the correlation between multiple independent factors and the dependent variables (threshold, discrimination, identification, and TDI).

## 4. Discussion

Intracranial tumours, including glioblastoma and astrocytic tumours, have identifiable genetic profiles that may influence their phenotype and have an impact on their clinical behaviour, therapeutic response, and overall survival time [22]. It is widely acknowledged that several genetic abnormalities that are observed in distinct cancer patients might result in the manifestation of a shared disease phenotype, hence posing challenges when elucidating the links between genotypes and phenotypes [23,24].

Genetic alterations and driving mutations of OGM exert an impact on their growth and behaviour. *SMO*, *AKT1*, and *PIK3CA* are often-observed somatic mutations. SMO-mutated OGMs are more common in older individuals, while *AKT1* mutations are more prevalent among younger populations.

Numerous studies have investigated the impact of driver mutations on the progression and features of OGM and have demonstrated that *SMO* and *AKT1* mutations enhance OGM prognostic assessment [13,25]. Nevertheless, the relationship between specific imaging features, genetic alterations in OGM, and preoperative olfactory function remains poorly understood.

This retrospective study aimed to identify the factors affecting preoperative olfactory function in patients with OGM. The analysis involved evaluating the molecular profiles, preoperative olfactory function, and imaging characteristics of the OGMs. The most common driver mutations in our study were found in *SMO/SUFU* and *AKT1* (Table 1). No significant differences in preoperative olfactory function were observed between *SMO/SUFU* or *AKT1* mutations and WT meningioma groups.

When categorized by mutation status, the median tumour volumes were 22 cm^3^ for *AKT1*, 22 cm^3^ for *SMO/SUFU*, and 42 cm^3^ for WT (*p* < 0.05). This suggests that there is a significant difference in tumour volume between the mutational groups, with WT patients having larger tumours on average in our study (Table 1). This finding contrasts with previous research indicating that olfactory groove meningiomas (OGMs) harbouring *SMO* mutations usually display a larger tumour volume when compared with those with other mutations or the wild type (WT) [10]. The exact cause of this phenomenon remains uncertain; however, aberrations in the *SMO* gene can result in anomalous stimulation of the Sonic Hedgehog (SHH) pathway, potentially causing heightened cellular proliferation and the development of tumours [10,25].

The spaciousness of the olfactory groove enables the tumour to grow without causing symptomatic compression on surrounding brain regions. As a result, OGMs may undergo a prolonged period of growth and reach a greater size before they are detected and treated.

Our study’s findings offer evidence supporting the correlation between tumour imaging criteria and the phenotypic characteristics of OGM, which are further associated with olfactory impairment.

Primarily, we observed that the presence of PSH seems to differ significantly across the mutation groups, with the lowest occurrence in the *AKT1* group and the highest in the *SMO* and WT groups.

Secondly, PSH was identified as an independent prognostic factor for preoperative reduced performance in odour threshold, discrimination, identification, and overall TDI scores. We have shown that 45% (R^2^ = 0.448) of the variation in TDI can be explained by the influence of PSH and OGM associated with perifocal oedema in this study. However, this means that 55% of the variation of olfactory function in OGM cases remains unexplained, necessitating further analysis to fully understand the relationship of tumour characteristics and morphology with TDI scores. Additionally, other factors, not included in our study, may also play a role in the determination of TDI scores [26].

The olfactory tracts, extensions of the olfactory bulbs, traverse the region of the planum sphenoidale. The tracts consist of second-order neurons that convey information from the olfactory bulbs to the brain. PSH in this region may potentially exert pressure on these neural pathways, resulting in impaired olfactory function. The specific damage to these structures would vary based on the degree and position of the hyperostosis, as well as the existence and development of any accompanying tumours [27].

Perifocal oedema—swelling that occurs around a tumour—is a common occurrence in brain tumours, can influence post-operative fatality, functional connectivity and significantly increases the incidence of preoperative symptoms, neurological deficits and postoperative complications [28,29,30].

In our cohort analysis, perifocal oedema emerged as a significant negative prognostic factor for odour identification and TDI.

Our data reveal that individuals with perifocal oedema have a higher median threshold score when compared with patients without. Thus, the results of the univariate and multivariate analysis regarding odour threshold and perifocal oedema are conflicting. This phenomenon may arise as a result of confounding variables or interactions between variables that are not readily observable when variables are analysed individually. This indicates that the interconnections among factors are intricate and that a comprehensive analysis of several variables is necessary for precise result prediction.

Perifocal oedema can result in symptoms such as headaches, nausea, and neurological impairments, which in turn can lead to the seeking of medical treatment and the need to undergo diagnostic testing. One could speculate that patients with perifocal oedema are more likely to be diagnosed promptly due to the symptoms it causes, though there is no supporting evidence for this in the literature.

Thirdly, within our study group, female patients outperformed male patients in preoperative odour identification and discrimination. However, in the multivariate analysis our data indicate that, in the context of this particular investigation, men exhibited superior olfactory discrimination (Table 3). Our findings reveal that gender explains 38% of the variation in odour discrimination. Although the data exhibit statistical significance, their predictive accuracy is constrained by their lower power (R^2^ = 0.384) (Table 3).

Multiple studies have indicated that women typically exhibit superior performance when compared with men in various domains of olfactory tests. Most of our OGM patients were female, which is consistent with the increased female preponderance in meningiomas. Hormonal influence on tumour growth has been discussed in previous studies [31,32]. Thus, although the concept connecting women’s olfactory superiority to the cellular composition of the olfactory bulb has been suggested, the current body of research presents contradictory results [33,34,35].

The Sniffin’ Sticks olfactory test battery exhibits a distinct correlation between performance and age in numerous studies [36,37,38]. Olfactory capabilities, encompassing the capacity to perceive, differentiate, and recognise scents, undergo substantial enhancement during childhood and adolescence, culminating in their zenith during early adulthood. From the age of 40 onwards, there is a steady decrease in the ability to detect smells. The decrease in cognitive function becomes more noticeable as individuals become older, particularly in those above the age of 71 who score much lower than younger persons. The connection between the sense of smell and cognitive functions is also confirmed by numerous studies on the development of various forms of dementia and olfactory disorders [39,40,41,42]. Regarding cognitive impairment associated with OGMs, Constanthin et al. conducted a study involving 17 patients and discovered that OGMs adversely affect cognitive processes related to the prefrontal cortex, specifically cognitive flexibility and attention [43]. From this, it can be deduced that age, tumour size, and frontal cognitive function impairment are potential confounding variables in olfactory performance for individuals with OGMs.

Olfactory function typically begins to decline around the age of 60 and intracranial meningiomas occur predominantly in populations over 65 years [36]. Although our *SMO/SUFU* mutant OGM patients were mostly >65 years old, our sample size may not be sufficient to confirm the predominance of meningiomas mutated by *AKT1* in younger patients and *SMO*/wild type in older patients [10].

The correlation between the genetic composition of a tumour and its behaviour and clinical manifestations is of utmost importance. Investigating this relationship could provide valuable insights into the origin, progression, and potential treatment options tailored to the genetic makeup of the tumour.

Our study identifies independent variables that predict olfactory performance in OGM patients, which can guide preoperative decision-making to accurately identify OGM patients at a high risk of postoperative olfactory impairment. While the olfactory system is highly complex and consists of several cell types that cooperate to recognize and interpret smells, we do not fully understand the mechanism of olfaction impairment in OGM patients. The investigation of potential molecular causes should include the examination of direct compression of the olfactory bulb, injury to olfactory receptor neurons, and disruption of the central olfactory pathways.

We did not identify any significant correlation between the OGM mutational status and olfactory performance because of the limited cohort; rather, we found an indirect relationship between the OGM mutational status and preoperative olfactory impairment. A notable disparity was observed in the OGM tumour volume and PSH when comparing the various mutational groups. Nevertheless, we successfully demonstrated a connection between the preoperative olfactory subtests and the PSH.

## 5. Study Limitations

Our study is subject to several limitations, of which the following are the most important:

The study was only able to analyse a small subgroup due to the lack of preoperative data on olfactory function in patients with OGM.The included molecular alterations have a naturally low frequency, making it difficult to interpret the results in terms of comprehensive statistical analyses and may not be representative of the larger population.Because of the small sample size, the study’s statistical power is diminished, making it harder to draw firm findings and thus rendering this more of an exploratory study.The study has a retrospective design.

## 6. Conclusions

Our research, involving 22 individuals diagnosed with olfactory groove meningiomas (OGM) sheds light on how the genetic makeup of these tumours influences their impact on the preoperative olfactory function. Despite the nature of the olfactory system and the gaps in our knowledge about how OGM affects smell perception, our study emphasizes the importance of considering factors in preoperative assessments. Though the small sample size limited our ability to establish a link between OGM mutations and olfactory function, we did observe a connection between preoperative smell tests, tumour volume and planum sphenoidale hyperostosis. These findings suggest that delving deeper into molecular aspects could offer insights that pave the way for personalized treatment strategies that focus on preserving smell function and improving the quality of life for OGM patients.

## Figures and Tables

**Table 1 cancers-16-01595-t001:** Women were over-represented (73%) and the majority of patients were >65 years old (59%). Odour subtests were assessed in 21 patients. Due to technical reasons, TDI was obtained in 22 patients. The *SMO/SUFU* group included the most anosmic patients (*n* = 6), followed by the *AKT1* group (*n* = 4). In cranial CT scans, PSH was detected in 100% of the *AKT1-/SMO*-WT group (*n* = 3), in 90% of the *SMO/SUFU* group (*n* = 10) and in 75% of the *AKT1*-group (*n* = 7) (*p* = 0.048). A significant disparity was observed when comparing the tumour volume of *SMO/SUFU* mutants with that of the *AKT1-/SMO*-WT tumours (*p* = 0.005).

Parameter		∑	*AKT1*	*SMO/SUFU*	*AKT1-/SMO*-Wild-Type (WT)	*p*-Value
All patients	(n (%))	22	7 (32)	10 (46)	5 (23)	
Sex	(n (%))					
Female		16 (73)	5	7	4	0.915 *
Male		6 (27)	2	3	1	
Age	(n (%))					
≤65 Years		9 (41)	2	4	3	0.549 *
>65 Years		13 (59)	5	6	2	
Tumour volume (median (range) cm^3^)		25 (2–48)	22 (4–45)	22 (2–39)	42 (28–48)	0.038 **
AKT1 vs. AKT1/SMO-WT						0.149 ***
SMO/SUFU vs. AKT1/SMO-WT						0.005 ***
Tumour volume	(n (%))					
≤25 cm^3^		9 (41)	4	5	---	0.102 *
>25 cm^3^		13 (59)	3	5	5	
Ethmoid cell infiltration	(n (%))					
Yes		6 (32)				
No		13 (68)				
PSH	(n (%))					
Yes		15 (75)	3	9	3	0.048 *
No		5 (25)	4	1	---	
Osseous-contrast enhancement [n (%)]						
Yes		5 (28)	2	2	1	0.961 *
No		13 (72)	5	6	2	
Perifocal oedema	(n (%))					
Yes		12 (71)	5	6	1	0.624 *
No		5 (29)	1	3	1	
Preoperative (Median (range))						
Threshold available		1.0 (0–10)	1.4 (1.0–4.3)	1.0 (0–10.0)	1.0 (0–1.3)	0.245 **
Discrimination available		7.0 (4–13)	8.5 (5.0–11.0)	7.0 (4.0–13.0)	6.0 (4.0–12.0)	0.494 **
Identification available		6.0 (2–16)	9.0 (5.0–14.0)	3.5 (2.0–16.0)	6.0 (3.0–10.0)	0.123 **
TDI		13.0 (6–35)	13.0 (11.5–28.8)	13.0 (6.0–34.8)	13.0 (8.0–22.0)	0.458 **
Olfactory performance, Sniffin’ Sticks						
Normosmia	(n (%))	2 (10)	1	1	---	0.844
Hyposmia		6 (30)	2	2	2	
Anosmia		12 (60)	4	6	2	

* Pearson’s chi-squared test; ** Kruskal–Wallis test; *** Mann–Whitney U test.

**Table 2 cancers-16-01595-t002:** For each grouping, the median score of the olfactory subtest, the range of scores, and the *p*-value of any statistically significant differences between categories are given. The table shows the patients’ gender, age, tumour volume, ethmoid cell infiltration, PSH, osseous enhancement, and perifocal oedema, as well as their demographic and clinical characteristics.

Threshold available, all patients (*n* = 21); 1 (1–10) (median (range))			
Gender	Female (*n* = 15)	Male (*n* = 6)	*p*-value
	1 (1–10)	1 (1–2)	0.381
Age	≤65 Years (*n* = 8)	>65 Years (*n* = 13)	
	1 (1–6)	1 (1–10)	0.677
WHO Grade	Grade 1 (*n* = 20)	Grade 2 (*n* = 1)	
	1 (1–10)		
Tumour volume	≤25 cm^3^ (*n* = 9)	>25 cm^3^ (*n* = 12)	
	2 (1–10)	1 (1–3.5)	0.181
Ethmoid cell infiltration	Yes (*n* = 6)	No (*n* = 12)	
	1 (1–10)	1 (1–6)	0.883
PSH	Yes (*n* = 15)	No (*n* = 4)	
	1 (1–10)	3 (1–6)	0.086
Osseous enhancement	Yes (*n* = 5)	No (*n* = 12)	
	1 (1–2)	1 (1–6)	0.823
Perifocal oedema	Yes (*n* = 5)	No (*n* = 11)	
	4 (1–10)	1 (1–4)	0.038
Discrimination available, all patients (*n* = 21); 7(4–13) (median (range))			
Gender	Female (*n* = 15)	Male (*n* = 6)	*p*-value
	8 (4–13)	5 (4–10)	0.055
Age	≤ 65 Years (*n* = 8)	>65 Years (*n* = 13)	
	6 (4–13)	8 (4–12)	0.145
WHO grade	Grade 1 (*n* = 20)	Grade 2 (*n* = 1)	
	7.5(4–13)		
Tumour volume	≤25 cm^3^ (*n* = 9)	>25 cm^3^ (*n* = 12)	
	8 (4–13)	7 (4–12)	0.591
Ethmoid cell infiltration	Yes (*n* = 6)	no (*n* = 12)	
	4 (4–10)	7.5 (5–13)	0.059
PSH	Yes (*n* = 15)	no (*n* = 4)	
	7 (4–10)	10.5 (5–13)	0.055
Osseous enhancement	Yes (*n* = 5)	no (*n* = 12)	
	4 (4–10)	7.5 (4–13)	0.151
Perifocal oedema	Yes (*n* = 5)	no (*n* = 11)	
	9 (4–13)	7 (4–11)	0.229
Identification available (*n* = 21); 6(2–16) (median (range))			
Gender	Female (*n* = 15)	Male (*n* = 6)	*p*-value
	6 (2–16)	3.5 (3–14)	0.326
Age	≤65 Years (*n* = 8)	>65 Years (*n* = 13)	
	4.5 (3–16)	7 (2–14)	0.273
WHO grade	Grade 1 (*n* = 20)	Grade 2 (*n* = 1)	
	6 (3–16)		
Tumour volume	≤25 cm^3^ (*n* = 9)	>25 cm^3^ (*n* = 12)	
	7 (2–16)	4.5 (3–10)	0.062
Ethmoid cell infiltration	Yes (*n* = 6)	No (*n* = 12)	
	3 (2–14)	6 (3–16)	0.154
PSH	Yes (*n* = 15)	No (*n* = 4)	
	4 (2–11)	14 (7–16)	0.006
Osseous enhancement	Yes (*n* = 5)	No (*n* = 12)	
	3 (3–14)	6 (2–16)	0.423
Perifocal oedema	Yes (*n* = 5)	No (*n* = 11)	
	8 (3–16)	4 (2–14)	0.134
TDI available (*n* = 22); 13(6–35) (median (Range))			
Gender	Female (*n* = 16)	Male (*n* = 6)	*p*-value
	14 (6–35)	9.5 (8–26)	0.109
Age	≤65 Years (*n* = 9)	>65 Years (*n* = 13)	
	11.5 (8–35)	15.5 (6–29)	0.105
WHO grade	Grade 1 (*n* = 20)	Grade 2 (*n* = 2)	
	13 (8–35)		
Tumour volume	≤25 cm^3^ (*n* = 9)	>25 cm^3^ (*n* = 13)	
	25 (6–35)	13 (8–22)	0.146
Ethmoid cell infiltration	Yes (*n* = 6)	No (*n* = 13)	
	8 (6–26)	13 (9–35)	0.121
PSH	Yes (*n* = 15)	No (*n* = 5)	
	13 (6–25)	26 (11.5–35)	0.058
Osseous enhancement	Yes (*n* = 5)	No (*n* = 13)	
	8 (8–26)	13 (6–35)	0.212
Perifocal oedema	Yes (*n* = 5)	No (*n* = 12)	
	25 (8–35)	13 (6–29)	0.136

**Table 3 cancers-16-01595-t003:** The table provides the estimated coefficients (in the second column), standard errors (in the third column), t values (in the fourth column), and *p* values (in the fifth column) for each of the independent variables. The sixth and seventh columns show the lower and upper bounds of the 95% confidence intervals for the coefficients, respectively. The last column provides the overall test statistics for the regression model, including the number of observations (*n*), the coefficient of determination (R^2^), the adjusted R^2^, the F statistic, and the *p*-value for the overall model.

**Multivariate Linear Regression Analysis: Independent Variables in All Models:** *AKT1*_Mut, *SMO_SUFU*, *AKT1-/SMO*-wild-type (WT), age 65, sex, volume (≤25/>25 cm^3^), ethmoid cell infiltration_(yes/no), PSH_yes (yes/no), osseous enhancement_yes (yes/no), perifocal oedema_yes (yes/no)							
**Threshold**						95.0% KI	
Coefficient	b	SE	β	T	*p*	LB	UB
PSH (no/yes)	−2.639	0.658	−0.508	−4.012	0.001	−4.061	−1.218
Osseous enhancement (no/yes)	−2.142	0.770	−0.389	−2.783	0.016	−3.805	−0.480
Age (≤65/>65 years)	−1.589	0.682	−0.329	−2.330	0.037	−3.062	−0.116
Comment *n* = 21; R^2^ = 0.864; Adj. R^2^ = 0.790; F (7, 13) = 11.749; *p* < 0.001							
**Discrimination**							
Sex (female/male)	−2.672	1.037	−0.453	−2.576	0.019	−4.851	−0.493
PSH (no/yes)	−3.078	1.037	−0.522	−2.967	0.008	−5.257	−0.899
Comment *n* = 21; R^2^ = 0.446; Adj. R^2^ = 0.384; F (2, 18) = 7.242; *p* = 0.005							
**Identification**							
PSH (no/yes)	−7.169	1.204	−0.818	−5.952	<0.001	−9.723	−4.616
Perifocal oedema (no/yes)	−3.264	1.239	−0.412	−2.634	0.018	−5.890	−0.637
Comment *n* = 21; R^2^ = 0.744; Adj. R^2^ = 0.680; F (4, 16) = 11.620; *p* < 0.001							
**TDI**							
PSH (no/yes)	−9.748	2.801	−0.598	−3.481	0.003	−15.632	−3.864
Perifocal oedema (no/yes)	−8.991	3.076	−0.590	−2.922	0.009	−15.454	−2.527
Comment *n* = 22; R^2^ = 0.527; Adj. R^2^ = 0.448; F (3, 18) = 6.679; *p* = 0.003							

## Data Availability

The data presented in this study are available on request from the corresponding author, D.P. Data are contained within the article.

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
