# Peer review of "Correlation of Molecular Status with Preoperative Olfactory Function in Olfactory Groove Meningioma"

_cancers, 2024, doi:10.3390/cancers16081595_

Round 1
Reviewer 1 Report
Comments and Suggestions for Authors
The study aims to examine the possible correlation between genomic alterations and pre- operative olfactory function in patients with olfactory groove meningioma (OGM), due to the frequent presence of olfactory impairment. They utilized next-generation sequencing to analyze samples from 22 individuals with OGM to detect driver mutations. Tumor morphology was assessed using preoperative imaging, whereas olfactory function was examined using Sniffin' Sticks. In this study of OGM patients, mutations were as follows: 10 with SMO/SUFU, 7 with 17 AKT1, and 5 as wild-type. Planum sphenoidale hyperostosis (PSH) was present in 75% of patients, showing significant variation by mutation (p = .048). Tumor volumes, averaging 25 cm³, significantly differed among groups. Planum sphenoidale hyperostosis negatively impacted olfaction, notably affecting odor threshold, discrimination, identification, and global olfactory performance score. Age over 65 and female gender were linked to lower thresholds and discrimination scores. The study highlights planum sphenoidale hyperostosis and perifocal edema's significant effect on olfactory function in OGM patients but finds no link between olfactory impairment and tumor mutations. It suggests age and gender affect olfactory impairment. This is an interesting study with limited clinical applications.
Comments on the Quality of English LanguageMinor editing might be helpful.
Author Response
Dear Reviewer 1; thank you very much for your suggestions regarding the minor English corrections needed for our medical article. Your expertise has been invaluable in enhancing the clarity and readability of our work.
I am pleased to inform you that we have thoroughly reviewed the article and implemented the necessary revisions. To ensure the level of accuracy, in language we enlisted the help of an English speaker who carefully reviewed and edited the document. This has significantly improved the flow and precision of the language used throughout the article. We are confident that these amendments have addressed your concerns and have elevated the quality of our publication. We appreciate your comments and are eager to present a professional document for your final review.
Reviewer 2 Report
Comments and Suggestions for Authors
The manuscript "Correlation of Molecular Status with Preoperative Olfactory Function in Olfactory Groove Meningioma" by D Podlesek and colleagues reports on the olfactory function of 22 patients (with a large bias for female patients 16F/6M) studied before surgery for an olfactory grove meningioma. Pathogenic variants and gene losses of the most common genes involved in this disease were characterized on the basis of a custom designed amplicon panel applied to DNA extracted by tissue macrodissected from formalin fixed paraffin embedded samples of the meningiomas. Full neuroradiological characterization of the lesions was also provided and used for multivariate analysis.
According to the authors' conclusion only planum sphenoidale hyperostosis and perifocal edema were significantly correlated with worst preoperatory olfactory function while specific genetic alterations were not significantly correlated with preoperatory olfactory function. The work is interesting since is trying to elucidate if there is a connection between known pathogenic variants and preoperative olfactory function in olfactory grove meningiomas. However, as recognized by the authors the relative small number of patients and its retrospective character cast a doubt on the general value of their findings. Despite this, it may serve to focus attention of neurologists, neurosurgeons and ENTs on this peculiar aspect of the meningiomas. I suggest to add to the discussion some considerations on another possible confounding factor that is the large age variation among their patients since it is know that olfaction decrease with age and cognitive impairment. Another parameter that should be explicitly considered and discussed is a possible link between symptoms of cognitive decline linked to meningioma and their performance on the Sniffin' Sticks olfactory test battery.
Minoir Points
pg 3 row 105 Please substitute amplicon for amplikon
Comments on the Quality of English LanguageFine
Author Response
Dear Reviewer 2:
Thank you very much for your time, your insightful comments and suggestions regarding our manuscript entitled "Correlation of Molecular Status with Preoperative Olfactory Function in Olfactory Groove Meningioma." Your feedback has been invaluable in enhancing the quality and clarity of our study. We acknowledge your concerns about the potential limitations of our study, including the relatively small patient cohort and its retrospective nature. We understand that these factors may affect the generalizability of our findings. In response to your suggestions, we have made the following revisions and additions to our manuscript:
Age Variation Consideration:
As you rightly pointed out, olfactory function is known to decline with age, and this could intersect with cognitive impairments.
Cognitive Decline and Olfactory Performance:
We have explicitly addressed the potential link between symptoms of cognitive decline associated with meningiomas and performance on the Sniffin' Sticks olfactory test battery. We have included a subsection in the discussion that explores the complex relationship between cognitive function and olfactory abilities, drawing on current literature to contextualize our findings within the broader scope of neurocognitive and olfactory research. (pg 11; row 303-317).
We believe that these revisions address your concerns and enrich the discussion of our findings. We hope that our manuscript now provides a more comprehensive understanding of the factors influencing olfactory function in OGM patients and serves as a valuable resource for neurologists, neurosurgeons, and ENT specialists. We appreciate the opportunity to improve our manuscript and thank you for your contribution to the peer-review process.
Minor Points
pg 3 row 105 Please substitute amplicon for amplikon - done
Reviewer 3 Report
Comments and Suggestions for Authors
Hypothesis: Previous studies have focused on the relationship between mutational status, tumour histology, recurrence, and outcomes in meningiomas; the present study aims to delineate subgroups of OGM patients based on the impact of mutational status and image morphology on preoperative olfactory performance. The authors hypothesize that the mutational status of OGM influences preoperative olfactory performance and is associated with distinct image characteristics.
The impact of genetic mutations on olfactory function in OGM remains unknown despite the well-documented influence of tumour genomics on location, recurrence potential, image characteristics, and outcomes of meningiomas. Subsequently, investigating molecular alterations and their correlation with psychophysical tests can provide valuable insights.
Tumor volumes, averaging 25 cm³, significantly differed among groups. PSH negatively impacted olfaction, notably affecting odour threshold, discrimination, identification, and global olfactory performance score (TDI). Perifocal oedema was associated with lower TDI and altered threshold scores. Age over 65 and female gender were linked to lower thresholds and discrimination scores.
Conclusions of the authors: The study highlighted PSH and perifocal oedema's significant effect on olfactory function in OGM patients but found no link between olfactory impairment and tumour mutations, possibly due to the small sample size. It suggests that age and gender affect olfactory impairment. Additional research with a larger group of participants was suggested to be able to explore the impact of OGM driver mutations on olfactory performance.
The authors have not shown any theoretical evidence, any reasoning in a clinical history to support that there may be a link between olfaction and the biological nature of the tumour. Compression of the nerve may explain impairment of the vascular supply to the nerve.
Why should tumour grading influence olfaction, and what could be behind it?
Author Response
Answer:
According to the literature, there are links between olfaction and the biological nature of the tumor:
- Strickland et al. have shown the association between SMO mutational status and the tumor volume. In anterior skull base meningioma patients, SMO mutated presented with larger tumors compared to AKT1 mutated and wild-type meningiomas. The majority of those occurred in the olfactory groove [1].
- Michaela Dedeciusova al. has shown that a higher meningioma volume is associated with worse olfactory function before and after the surgery and that the preoperative normosmia is the most important prognostic factor for functional olfactory outcome [2].
- Welge-Luessen et. al. discusses how the growth patterns and location of OGM´s, can affect olfactory pathways and lead to symptoms like anosmia [3].
- Patients with larger lesions, which could be indicative of more aggressive or advanced biological behavior, present with olfactory dysfunction more frequently than those with smaller tumors [4].
- Also, the biological environment of the olfactory system, which could be disrupted by the presence of OGMs, can be influenced by more aggressive tumors [5].
- Here are some considerations and suggestions on why and how tumor grading might impact olfactory function:
- Impact of tumor size on mass effect:
Meningiomas of a higher grade tend to exhibit a faster growth rate and attain larger dimensions compared to tumors of a lower grade. A higher tumor size can have a more significant impact on the olfactory bulb and tract. The pressure can influence the vessels that are responsible for the supply of olfactory structures or, as a steal effect, those structures can be chronically underperfussed.
- Tumor invasiveness:
Higher grade meningiomas have a greater tendency to invade adjacent brain tissue, perhaps causing more significant damage to the olfactory bulb and nearby areas compared to lower-grade tumors. This intrusive behaviour can result in direct harm to the olfactory pathways or interfere with the blood flow to these regions, causing additional impairment to the sense of smell.
2. SMO/AKT1 Mutations:
As already mentioned above mutations in the SMO gene in meningiomas can cause inappropriate activation of the Hedgehog signaling system. This activation can lead to tumor development and aggressiveness. As a result, it may indirectly impair olfactory function by increasing the mass effect on surrounding tissue or invading olfactory structures. Although the precise effects of SMO mutations on the olfaction in OGMs are not completely comprehended, the use of Hedgehog pathway inhibitors to target these mutations may have an impact on tumor growth and can aid in the preservation of the sense of smell. AKT1 mutations, which are crucial in the PI3K/AKT/mTOR signaling pathway, are linked to lower-grade meningiomas and may display less aggressive behavior, indicating a less severe effect on olfactory function. These mutations may also impact the response to targeted medicines that block the PI3K/AKT/mTOR pathway, thereby changing olfactory results.
Although additional research is needed to fully understand the specific effects of SMO and AKT1 mutations on olfaction in OGMs, it is clear that these genetic changes might impact tumor features and treatment responses, which in turn may have an indirect effect on olfactory outcomes. The investigations into these mutations will bolster the comprehension of OGM pathophysiology and direct the advancement of focused therapeutics.
- Peritumoral edema:
Tumors of a higher grade often result in swelling and inflammation surrounding the tumor. This can have an impact on the adjacent brain tissue and cause increased pressure, as well as disrupt the normal signaling pathways that are crucial for the sense of smell.
We think that understanding how tumor grade influences olfaction in patients with olfactory groove meningiomas (OGMs) is crucial for patient management, outcome prediction, and developing strategies to mitigate olfactory dysfunction. Investigating the molecular processes, including the role of inflammatory markers, vascular alterations, and the direct impact on brain pathways, can offer insights into potential treatment targets to preserve or restore the sense of smell. Further research in this area is essential and has the potential to significantly enhance the comprehensive care of patients with OGMs.
- Strickland MR, Gill CM, Nayyar N, D'Andrea MR, Thiede C, Juratli TA, Schackert G, Borger DR, Santagata S, Frosch MP, Cahill DP, Brastianos PK, Barker FG, 2nd (2017) Targeted sequencing of SMO and AKT1 in anterior skull base meningiomas. J Neurosurg 127: 438-444 doi:10.3171/2016.8.JNS161076
- Dedeciusova M, Svoboda N, Benes V, Astl J, Netuka D (2020) Olfaction in Olfactory Groove Meningiomas. J Neurol Surg A Cent Eur Neurosurg 81: 310-317 doi:10.1055/s-0040-1709165
- Welge-Luessen A, Temmel A, Quint C, Moll B, Wolf S, Hummel T (2001) Olfactory function in patients with olfactory groove meningioma. J Neurol Neurosurg Psychiatry 70: 218-221 doi:10.1136/jnnp.70.2.218
- Ung TH, Yang A, Aref M, Folzenlogen Z, Ramakrishnan V, Youssef AS (2019) Preservation of olfaction in anterior midline skull base meningiomas: a comprehensive approach. Acta Neurochir (Wien) 161: 729-735 doi:10.1007/s00701-019-03821-8
- Murtaza M, Chacko A, Delbaz A, Reshamwala R, Rayfield A, McMonagle B, St John JA, Ekberg JAK (2019) Why are olfactory ensheathing cell tumors so rare? Cancer Cell Int 19: 260 doi:10.1186/s12935-019-0989-5
Round 2
Reviewer 3 Report
Comments and Suggestions for Authors
The authors have proved that the prognosis of the olfaction deficit depends on the volume of the meningioma to be removed. The measure of meningioma has a correlation with its genomic alterations. The correlation between olfaction deficit and genomic alterations of meningiomas is indirect.
Author Response
Thank you for your comments.
Those are justified and we have changed the paragraph on page 10; line Nr. 338-340 according.